# Current and New Predictors for Treatment Response in Metastatic Colorectal Cancer. The Role of Circulating miRNAs as Biomarkers

**DOI:** 10.3390/ijms21062089

**Published:** 2020-03-18

**Authors:** Alexandra Gherman, Loredana Balacescu, Sinziana Gheorghe-Cetean, Catalin Vlad, Ovidiu Balacescu, Alexandru Irimie, Cosmin Lisencu

**Affiliations:** 111th Department of Medical Oncology, University of Medicine and Pharmacy “Iuliu Hatieganu”, 34–36 Republicii Street, 400015 Cluj-Napoca, Romania; 2Department of Functional Genomics, Proteomics and Experimental Pathology, The Oncology Institute “Prof. Dr. Ion Chiricuta”, 34–36 Republicii Street, 400015 Cluj-Napoca, Romania; 3Department of General and Inorganic Chemistry, “Iuliu Hatieganu” University of Medicine and Pharmacy, 4 Louis Pasteur Street, 400012 Cluj-Napoca, Romania; cetean.sinziana@umfcluj.ro; 411th Department of Oncological Surgery and Gynecological Oncology, “IuliuHatieganu” University of Medicine and Pharmacy, 34–36 Republicii Street, 400015 Cluj-Napoca, Romania; airimie@umfcluj.ro (A.I.);; 5Department of Surgery, The Oncology Institute “Prof. Dr. Ion Chiricuta”, 34–36 Republicii Street, 400015 Cluj-Napoca, Romania

**Keywords:** chemotherapy, prediction, treatment response, biomarkers, microRNA, colorectal cancer

## Abstract

Colorectal cancer (CRC) is the third most frequently diagnosed cancer in the world. More than half of all CRC patients will eventually develop metastases and require treatment accordingly, but few validated predictive factors for response to systemic treatments exist. In order to ascertain which patients benefit from specific treatments, there is a strong need for new and reliable biomarkers. We conducted a comprehensive search using the PUBMED database, up to December 2019, in order to identify relevant studies on predictive biomarkers for treatment response in metastatic CRC. We will herein present the currently used and potential biomarkers for treatment response and bring up-to-date knowledge on the role of circulating microRNAs, associated with chemotherapy and targeted therapy regimens used in metastatic CRC treatment. Molecular, tumor-related, disease-related, clinical, and laboratory predictive markers for treatment response were identified, mostly proposed, with few validated. Several circulating microRNAs have already proven their role of prediction for treatment response in CRC, but future clinical studies are needed to confirm their role as biomarkers across large cohorts of patients.

## 1. Introduction

Colorectal cancer (CRC) represents a major public health concern as it is the third most frequently diagnosed cancer and the fourth cause of cancer-related mortality worldwide [1]. The five-year survival rate is around 64.9% for all stages, while in metastatic cases, it only reaches 13.1% [2]. A big issue of CRC reveals that 25% of the patients are diagnosed with stage IV disease and 50% of all CRC patients will develop metastases during their disease [3]. The “continuum of care” approach comprising classical chemotherapy, molecularly targeted treatments, and metastases resection where feasible, has led to a median overall survival (OS) of 30 months in the metastatic setting. Unfortunately, despite the multidisciplinary approach and favorable therapeutic results, there is a considerable percentage of patients with inadequate response to treatments and a dismal prognosis. Currently, there are few validated predictive factors for chemotherapy response in metastatic colorectal cancer (mCRC). Most of the predictive factors for treatment response are still not validated by prospective clinical trials and are, in some cases, disappointing, while in highly treated patients, the emergence of resistant clones is a non-negligible reason for therapeutic failure.

Nevertheless, the potential of minimally-invasive biomarkers, such as those from blood, is still not validated. Due to their crucial role in cancer progression, but also to their high stability in blood, microRNAs (miRNAs) have shown great potential as new biomarkers for cancer detection, prognosis, and treatment response [4]. Much research has focused on identifying of specific tissue miRNAs that could serve as biomarkers for treatment response, specific for each chemotherapy drug, targeted treatment, or immunotherapy. A recent review article [5] focused on the value of circulating miRNAs as emerging biomarkers for the diagnosis, prognosis, and response-prediction in mCRC. However, identifying circulating miRNAs as minimally invasive biomarkers associated with CRC progression and its treatment response still represents a big challenge.

Our work intends to answer one of the questions asked by any clinician involved in the management of mCRC patients: what biomarkers of efficacy of systemic treatments are currently available and what are the perspectives for future development in this field, with focus on circulating miRNAs. Other well-documented works on this subject are available, such as the review article published by Taieb et al., [6] that covers the enzymatic and molecular biomarkers useful in clinical practice and perspectives of research in this area, in the light of the international oncological societies’ current recommendations. However, not properly validated, available and inexpensive predictive biomarkers for response such as the baseline clinical features, disease characteristics, and elements of the basic blood works, as well as tumor markers, together with treatment toxicities, should be added to more expensive enzymatic or molecular assays.

This article aimed to present up-to-date information concerning the current predictive factors for systemic therapy response in mCRC and the role of circulating miRNAs as predictors for response to systemic therapy in CRC, but also information about the miRNAs’ challenges in therapeutic use.

## 2. Methods

We conducted a literature search on the PUBMED (https://www.ncbi.nlm.nih.gov/pubmed/) database, in order to identify relevant studies published up to December 2019, using Medical Subject Headings (MeSH) and keywords. Firstly, different string searches were performed to identify current predictive markers of response to both chemotherapy regimens and targeted treatments considered standard treatment for mCRC; we only included studies performed in stage IV disease. Secondly, we performed a search of circulating miRNAs with predictive value for treatment response in CRC; we included studies regardless the stage of the disease, due to limited research studies for stage IV-only in this area. Search terms were: metastatic colorectal cancer, mCRC, advanced colorectal cancer, colorectal cancer, prediction, efficacy, response, resistance, biomarkers, chemotherapy, targeted treatment, immunotherapy, systemic treatment, 5-Fluorouracil, Capecitabine, Oxaliplatin, Irinotecan, trifluridine/tipiracil, Bevacizumab, anti-EGFRs, Cetuximab, Panitumumab, Aflibercept, Ramucirumab, Regorafenib, Larotrectinib/Entrectinib, Pembrolizumab, Nivolumab, Ipilimumab, circulating miRNAs, serum miRNAs, plasma miRNAs, whole blood miRNAs, exosomes. We also checked the references of the selected studies in order to further identify papers of interest.

All relevant papers were included: clinical trials, meta-analyses, pooled analyses, clinical studies, multicenter studies, observational studies, systematic reviews, treatment guidelines. We excluded experimental studies performed only on in vitro or in vivo models. Where available, predictivity indicators were described.

## 3. Overview of Systemic Therapies Used in Metastatic Colorectal Cancer

Standard chemotherapy for mCRC is based on fluoropyrimidines (5-Fluorouracil (5-FU), capecitabine), Oxaliplatin, Irinotecan and Trifluridine/tipiracil, the latter only in case of refractory disease. 5-FU can be delivered with either Oxaliplatin or Irinotecan, a combination of both or administered as monotherapy in selected patients. Pivotal studies in mCRC treatment have shown that, while mono-chemotherapy with fluoropyrimidines (5-FU, capecitabine) offered response rates (RR) of approximately 20–25% and a median OS of 12 months, by adding a second agent, such as Irinotecan or Oxaliplatin, a doubled RR, and a prolonged survival rate are achieved [7,8]. However, a recent Cochrane review [9] questions the OS benefit attributable to the addition of a fluoropyrimidine to Irinotecan over Irinotecan alone in first- or second-line settings in mCRC.

The sequential administration of combined regimens based on 5-FU + Irinotecan (FOLFIRI) followed by 5-FU + Oxaliplatin (FOLFOX) or vice versa, leads to a median survival of 21 months [10]. However, the triple combination of 5-FU, Irinotecan, and Oxaliplatin proved its superiority to bi-therapies in an Italian clinical trial in terms of response rates: 66% for the triplet vs. 41% for 5-FU + Irinotecan, also with a survival advantage [11], results that were not confirmed by a similarly designed Greek study [12]. The addition of molecular targeted treatments to classical chemotherapy or as monotherapy in different treatment lines for metastatic disease, achieved significant improvements in therapy outcomes. The sequential administration of chemotherapy and targeted therapy regimens leads to an OS of up to 30 months for patients treated with a “continuum of care” perspective [13].

Targeted therapies approved and most commonly used in mCRC are the epidermal growth factor receptors (EGFR) inhibitors in Rast Sarcoma RAS genes wild-type disease: Cetuximab, Panitumumab, the antiangiogenics: Bevacizumab, Aflibercept, Ramucirumab, Regorafenib. Larotrectinib or Entrectinib are indicated in any refractory, solid tumor that exhibits a neurotrophic tyrosine receptor kinase (NTRK) gene fusion. Immunotherapy in mCRC is approved in the US, but not in Europe and is recommended in microsatellite instability high (MSI-H) refractory tumors.

The determinants in choosing the type and sequence of treatments in stage IV disease are related to patient (age, performance status, comorbidities, personal option), disease (symptoms, resectability, tumor biology, tumor burden, clinical evolution) and therapy itself (efficacy, toxicities, availability, costs), guidelines and approvals. Considering these, it has become more and more evident that an improvement of the therapy outcome has to be based on personalized features, including current approaches, but also new, specific biomarkers.

We will present herein the up to date information about the known validated and potential markers related to general treatment efficacy in metastatic disease, but also for each chemotherapy, targeted treatment drug and immunotherapy (Table 1), considered standard treatment according to the international guidelines of mCRC, as follows.

## 4. Predictive Factors for Chemotherapy Response in Metastatic Colorectal Cancer

Previous data pointed out that there is no consistent data to support the advanced age, male sex, and high body mass index as negative predictive factors [14]. Regarding the performance status (PS), a pooled analysis on outcomes of mCRC patients included in nine first-line treatment trials showed that patients with a PS=2 showed similar clinical outcomes compared to PS=0–1, but increased toxicities and 60-day mortality [15]. According to Schmoll et al. [16], symptomatic peritoneal carcinomatosis and multiple-site metastatic disease are negative predictive factors, while regressing carcinoembrionary antigen (CEA) dynamics through treatment is a positive predictive factor.

5-FU inhibits the thymidylate synthase (TS), a crucial enzyme for DNA synthesis. An adequate inhibition of TS leads to chemosensitivity to 5-FU, while high TS levels and/or its gene polymorphisms (TSER*3/TSER*3) may be involved in 5-FU resistance, but its use is not recommended in clinical practice [17]. Several studies have shown discordant levels of TS between the primary tumor and metastases, which explains the different clinical responses to 5-FU [18]. Metzger et al. [19], showed that a high basal level of thymidine phosphorylase (TP) in CRC is associated with lack of response to 5-FU and vice versa. Salonga et al. [20] found that patients expressing low levels of TS, TP and dihydropyrimidine dehydrogenase (DPD) had better responses than patients that had low levels of only one of the enzymes, but their use is not recommended with this purpose. Kohne et al. [21] studied the data from 3825 mCRC patients treated within 19 randomized prospective trials with a 5-FU-based regimen and found low PS (≥2), high number of involved metastatic sites (≥2) and peritoneal or liver sites of metastases to be predictive factors for resistance to chemotherapy. On the other hand, the presence of a rectal primary, lung, or nodal metastases predicted better outcomes in multivariate analysis. Among the laboratory parameters: high levels of white blood cells count (WBC) (≥10 × 10^9^/L), high platelets (≥400 × 10^9^/L), low hemoglobin (<11 g/dL), high alkaline phosphatase (>300 U/L) were associated with worse responses.

Capecitabine is a 5-FU prodrug. Low levels of TS or certain TS promotor polymorphisms (homozygous for the genotype S/S versus S/L or L/L) [22], as well as low levels of DPD [23], seem to be associated with good responses. The occurrence of hand-food syndrome during treatment is considered to be a predictive marker of efficacy [13].

Irinotecan exerts a cytotoxic action by inhibiting type 1 topoisomerase (TOPO 1). High gene expression of TOPO 1 enhances the response to Irinotecan and may be a predictor of Irinotecan responsiveness, as a biomarker analysis on 1313 people enrolled in the FOCUS trial showed [24]. Freyer et al. [25] performed a biomarker analysis on 455 mCRC patients enrolled in four clinical trials after tumor progression on first line 5-FU. Their data pointed out normal baseline hemoglobin level, time since diagnosis shorter than nine months, oligometastatic disease (one organ involved), grade 3/4 (G3/G4) neutropenia and diarrhea at first cycle, as predictive factors for tumor response. Another positive predictive factor for Irinotecan response seems to be PS 0–1 [8].

Oxaliplatin (L-OHP) exerts synergistic effects with fluoropyrimidines and has shown a wide range of RR of 28–65% in the metastatic setting, either in first- or second- line settings. DNA excision repair protein 1 (ERCC-1), ERCC-2 gene polymorphisms, X-ray repair cross-complementing protein 1 (XRCC-1) polymorphisms seem to be associated with tumor response. Amongst these, the most studied is ERCC-1: high level of its expression is predictive of a poor response to Oxaliplatin chemotherapy, but this still needs to be validated in prospective clinical trials [12]. According to Shirota et al. [26], low intratumorally ERCC1 and TS are predictive for tumor response in advanced CRC. Potential biomarkers for lack of treatment response are PS ≥ 2, elevated number of prior chemotherapy regimens (≥3), low baseline hemoglobin level (<10 g/dL) and frequency of administration (triweekly worse than biweekly), according to a study on 481 mCRC patients, 5-FU resistant [27].

Trifluridin/tipiracil is an oral combination drug approved in 2015 in the US and in 2016 in the EU as a third- or fourth-line treatment of mCRC. There are currently no validated biomarkers for response prediction to this drug, although it was suggested that neutropenia after first administration could be an indicator of response [28].

## 5. Predictive Factors for Response to Targeted Treatments and Immunotherapy in Metastatic Colorectal Cancer

Monoclonal Antibody Therapies Targeting EGFR (Cetuximab, Panitumumab)**:** About 40%–80% of CRC tumors present an over-expression of the EGFR [29]. Clinical trials failed to show a predictive role of the EGFR gene status for the treatment with EGFR inhibitors [30]. Responses are seen in no more than 30% of an unselected patient population [31]. They are administered in combination with FOLFOX (5-FU + folinic acid + Oxaliplatin) or FOLFIRI (5-FU + folinic acid + Irinotecan) chemotherapy in the first-line setting, associated with Irinotecan-based chemotherapy in any treatment line or as monotherapy after failure of Irinotecan and Oxaliplatin-based chemotherapy or in case of intolerance to Irinotecan [13].

Currently validated biomarkers of efficacy are: RAS gene mutational status (mutations in the KRAS exons 2, 3, 4-codons 12, 13, 59, 61, 117, 146 and NRAS exons 2, 3, 4-codons 12, 13, 59, 61, 117) and tumor sidedness [13], their use being restricted to RAS wild-type patients [31] and left-sided tumors, as tumors located in the right colon have a more aggressive course, worse prognosis [32], and a lack of response to anti-EGFR agents [33].

BRAF mutations (V600E in the vast majority of cases) are present in up to 8%–12% of mCRC patients and in two-thirds of the cases the tumors are located on the right side of the colon. The BRAF mutation in mCRC implies a significantly negative prognosis [34], but not in MSI-H tumors [35]. In the second and further line settings, the role of BRAF mutations as markers of resistance to EGFR inhibitors is more evident than in first-line settings, where the results reported are controversial [36].

Other proposed positive predictive factors for response to EGFR inhibitors are the development of skin toxicity during the treatment and hypomagnesemia [37]. Proposed markers of resistance are: amplification of KRAS proto-oncogene, GTPase (KRAS), Erb-b2 receptor tyrosine kinase 2 (HER2) and MET proto-oncogene, receptor tyrosine kinase (MET), phosphatidylinositol-4,5-bisphosphate 3-kinase catalytic subunit alpha (PIK3CA) exon 20 mutation, alteration of phosphatase and tensin homolog (PTEN) [13], increased transforming growth factor TGF alpha [38], amphiregulin (AREG) and epiregulin (EREG) suppression [39].

Bevacizumab: There are no validated predictive biomarkers for Bevacizumab efficacy [12]. The potential predictive role of germline polymorphisms of the vascular endothelial growth factor (VEGF) genes on the Bevacizumab treatment response is not yet established. The study performed by Formica et al. [40], showed that the VEGF gene polymorphism 1154 (G/G over G/A + A/A) was predictive for better progression-free survival (PFS) and polymorphism 634 was predictive for a better response (G/G vs. G/C + C/C). The occurrence of treatment-induced arterial hypertension seems to be associated with better outcomes after Bevacizumab treatment with regards to OS, PFS and RR [41]. A prospective, randomized, multicenter study (ITACa) on 289 mCRC patients showed that low systemic immune inflammation (SII) indices, especially low neutrophil-to-lymphocyte ratio (NLR) and low platelet-to-lymphocyte ratio (PLR) are suitable prognostic and predictive biomarkers in patients receiving chemotherapy plus Bevacizumab [42]. Formica et al. [43] performed an analysis of 87 treatment-naive mCRC patients who had baseline and biweekly measurements of CEA and carbohydrate antigen 19-9 (CA 19-9) during their first-line treatment with chemotherapy +/− Bevacizumab and showed that patients with abnormal baseline CA 19-9 benefited significantly from the combination treatment.

Aflibercept is an antiangiogenic drug approved for administration in combination with FOLFIRI chemotherapy, after disease progression under first-line Oxaliplatin-based treatment in mCRC. An analysis of patients included in the phase II AFFIRM clinical trial showed that elevated baseline plasmatic levels of IL-8 and subsequent increases during the treatment were associated with worse PFS in patients treated with mFOLFOX6 + Aflibercept in first-line treatment of mCRC [44].

Ramucirumab is a VEGFR2 - binding monoclonal antibody used in second-line treatment of mCRC that improves both PFS and OS, in combination with FOLFIRI. Data related to biomarker results in the RAISE study were available for 894 patients and showed that a high plasma level of VEGFD (>115 pg/mL) is a potential predictive biomarker for treatment response [45].

Regorafenib is an antiangiogenic multi-kinase inhibitor used in mCRC for refractory tumors, that inhibits the VEGFR2-TIE2 tyrosine kinase pathway. There are not yet validated markers to predict tumor response to this drug. However, hand-foot skin reaction and lung nodule cavitation have been recently reported as potential clinical biomarkers for treatment response [46].

Larotrectinib and Entrectinib are targeted therapies approved for the treatment of refractory advanced or metastatic solid tumors, including CRC, that exhibit a neurotrophic tyrosine receptor kinase (NTRK) gene fusion, without other treatment options. The NTRK gene fusions are present in approximately 0.2% to 1% of CRC [47]. No predictive factors for response are currently known for these drugs among the carriers of the gene fusion.

Immunotherapy: Pembrolizumab and the combination of Nivolumab + Ipilimumab are FDA-approved in progressive unresectable/metastatic colorectal cancer displaying MSI-H status or deficient mismatch repair genes (dMMR) without other valid treatment options. A clinical study on Pembrolizumab showed that MMR tumors were associated with immune-related RR of 40% and six-months PFS of 78%, while proficient MMR showed RR of 0% and six-months PFS of 11% [48].

## 6. The Role of the miRNAs in Predicting the Treatment Response in Colorectal Cancer

### 6.1. A Short Overview about miRNAs

MiRNAs are small, non-coding RNAs of 18–24 nucleotides that play a crucial role in controlling gene expression at the post-transcriptional level. However, in cancers, due to alteration of their levels, miRNAs are involved in and influence the majority of cancer hallmarks [49]. Depending on their target genes, miRNAs are defined as oncomiR when targeting tumor-suppressor mRNA, and tumor-suppressor miRNA (TSmiRNA) when targeting oncogenes [50].

Regarding their biogenesis, miRNAs are firstly transcribed and preprocessed in the nucleus, then exported to the cytoplasm where are processed to mature miRNAs in association with RISC (RNA-induced silencing complex). Functionally mature miRNAs bind complementary regions of mRNA targets, inducing either inhibition of translation or mRNA degradation. While the majority of miRNAs are identified in the cellular environment, there are many miRNAs identified both in the tumor microenvironment and blood (serum and plasma), including exosomes [4].

One of the most significant advantages of circulating extracellular miRNAs as biomarkers is due to their high stability during blood processing, providing reliable molecular data even from a low volume of serum/plasma or a reduced number of exosomes.

### 6.2. MiRNAs Predicting Treatment Response in CRC

Increasing evidence has suggested that both tissular and cell-free miRNAs represent useful tools for cancer management even if future multicenter studies still have to validate their role as tumor biomarkers before being implemented in clinical practice. In a previous study, Dong et al. [51] investigated the role of miR-429 in the diagnosis, prognosis and response prediction to first-line, 5-FU-based chemotherapy in CRC patients with various clinical stages (I-IV), according to the seventh edition of the TNM Classification of Malignant Tumors. The diagnostic and prognostic roles of miR-429 were first proved on tissue samples: an analysis on 78 pairs of CRC tissues and adjacent healthy tissues showed that miR-429 expression was significantly higher in the cancerous tissues versus normal ones and it correlated with the TNM stage, lymph node metastasis, primary tumor size, but also, higher expression of miR-429 signified a worse OS. Further, to investigate the role of miR-429 as a putative serum biomarker, they analyzed its expression on 45 serum samples of CRC patients collected preoperatively (stages I-IV) and 45 serum samples from healthy donors. A significantly higher miR-429 expression was found in CRC patients versus healthy donors and it also correlated with TNM stage.

Moreover, this study was extended on a different set of 116 mCRC patients that received first-line 5-FU based chemotherapy, to explore the tissue levels of miR-429 with regards to chemotherapy response according to Response Evaluation Criteria in Solid Tumors version 1.0 (RECIST v1.0) criteria. Their data showed that the responders (complete response + partial response) had significantly lower levels of miR-429, with no differences between the type of 5 FU-based protocol administered (FOLFIRI, FOLFOX, 5-FU+folinic acid) than non-responders (stable disease + progressive disease). Furthermore, on multivariate analysis, miR-429 was an independent prognostic factor for 5-FU-based first-line chemotherapy response. ROC-curve analysis revealed an area under the curve (AUC) for miR-429 of 0.721 (95% CI: 0.630–0.800) while the diagnostic sensitivity and specificity were 52.70% and 85.71%, respectively, certifying that miR-429 can be associated with 5-FU-based chemotherapy response.

In another study, Ren D et al. [52] tested the hypothesis that oncomiR-196b-5p promotes stemness and, therefore, chemoresistance through the JAK-STAT3 pathway. Firstly, the diagnostic role of miR-196b-5p was evaluated by analyzing its levels on 20 pairs of CRC tissues and normal adjacent ones and found that in 19/20 cases, it was higher in primary CRC tissues vs. healthy surrounding tissues. Next, this group investigated the prognostic value of miR-196b-5p on a different set of 90 CRC tissues. The authors established a significantly positive correlation between increased expression of miR-196b-5p and the presence of metastatic disease. Moreover, lower OS was observed for patients with increased expression of miR-196b-5p compared with those with low levels of miR-196b-5p. Further on, investigating serum samples of 150 CRC patients and 90 healthy donors, the authors showed that miR-196b-5p expression in both serum and serum-derived exosomes was significantly elevated in CRC patients compared to the control group and it was positively correlated with T and M stage. Regarding chemosensitivity, in vitro studies showed that overexpression of miR-196b-5p activates STAT 3 signaling pathway by directly targeting SOCS 1 and 3, promoting the chemoresistance and stemness of CRC cells, while silencing oncomiR-196b-5p sensitizes them to 5-FU *in vivo.*

In an attempt to identify CRC serum miRNA-based biomarkers, Qin Y et al. [53], performed a study in order to demonstrate that serum miR-135b is overexpressed in CRC patients and that is involved in L-OHP chemoresistance. They used the serum samples of 25 CRC patients and 25 healthy donors and showed that miR-135b was significantly overexpressed in CRC patients versus healthy donors. Regarding chemosensitivity, knockdown of miR-135b sensitized CRC cells to L-OHP, so the authors proposed the analysis of an antimiR-135b that could potentially sensitize CRC cells to L-OHP by increasing the expression of FOXO 1; the miR-135b/FOXO1 axis promotes mitochondrial apoptosis in colorectal cancer cells treated with L-OHP. Moreover, an in vivo investigation on transfected mice revealed that antimiR-135b intensified the antitumoral effect of L-OHP, highlighting the role of miR-135b in oxaliplatin chemoresistance.

Previous studies have shown that miR-143 is downregulated in cancer. Qian X et al. [54], tested the role of miR-143 as both diagnosis and treatment biomarkers in CRC. Firstly, they included 62 pairs of CRC tissues and adjacent healthy tissues and found that miR-143 was significantly downregulated in tumors vs. healthy tissues. Moreover, comparing the tumor stages, they observed that advanced stages (Dukes C+D) present a lower expression of miR-143 than early stages (Dukes A+B). Considering insulin-like growth factor-I receptor (IGF-IR) is a target of miR-143, they also investigated the relationship between these molecules and concluded that miR-143 directly inhibits IGF-IR’s expression in vivo. Regarding its effects on the chemotherapy response, the authors demonstrated in in vitro study that overexpression of miR-143 significantly enhanced chemosensitivity to L-OHP through caspase 3, while overexpression of IGF1R exerted opposite effects. The plasmatic levels of miR-143 were significantly lower in CRC patients (*n* = 41) compared to healthy subjects (*n* = 10), meaning that its plasmatic levels can be considered as noninvasive biomarkers of diagnosis and possible for drug resistance.

In the same direction, miR-378 has been reported to be frequently downregulated in both CRC tissues and cell lines, but with discordant, elevated serum levels. In this regard, Wang et al. [55], tested the expression level of miR-378 in 20 pairs of CRC tissues (primary or metastatic) and healthy adjacent tissues from patients with CRC that underwent surgery. Their data pointed out a downregulation of miR-378 in 18/20 CRC tissues vs. healthy ones. When the levels of miR-378 were investigated in serum samples from 20 primary CRC patients, 17 mCRC patients and 14 healthy donors, elevated serum levels of miR-378 were observed in all CRC patients (primary or metastatic) vs. healthy donors, but no correlation with clinical and pathological features and the clinical outcome could be established. Based on the in vitro approach, CDC40 was identified as a potential target of miR-378, and high expression of miR-378 leads to L-OHP-induced apoptosis. In vivo studies on mice transplanted with miR-378 mimics and miR-378 control transfected CRC cells demonstrated that tumor cells with hyperexpression of miR-378 were smaller than the control group. Taking into account the above-mentioned data, the authors’ conclusion was that miR-378 could be used as a prognostic and predictive factor for chemoresistance.

The association of miRNAs predictive response with FOLFOX and FOLFIRI regimens has been also investigated. In this way, Chen Q et al. [56], evaluated the predictive potential of serum miRNAs in response to 1^st^ line FOLFOX in advanced CRC patients. The patients were separated depending on their response to the FOLFOX regimen: complete response, partial response, stable disease, and progressive disease. A miRNAs microarray assessment on eight serum samples from the response phase and eight from the resistance phase patients revealed 62 statistically different expressed miRNAs. Five out of these miRNAs (miR-221, miR-222, miR-122, miR-19a and miR-144) had a minimum 4.5-fold expression and were selected for validation in a larger population, included 36 response-phase patients and 36 resistance-phase patients. The validation data showed that the expression of serum miR-19a was significantly upregulated in the resistance phase versus the response phase, with no significant differential expression in the other four miRs (sensitivity 66.7%, specificity 63.9%). However, no significant differences were identified between the intrinsic resistance (PD at the first evaluation during FOLFOX) and acquired resistance (PD at further response evaluations).

The prognostic and predictive value of serum exosomal miRNAs on the clinical outcome of stage II-III CRC patients treated with adjuvant chemotherapy was investigated by Liu et al. [57]. They included 84 blood samples, prospectively collected and provided RNA sequencing. MiRNA analysis was reported to the patients with or without recurrence. Bioinformatics analysis and data validation revealed that exosomal miR-4772-3p could be a prognostic biomarker for tumor recurrence in stage II and III CRC patients. Patients with lower levels of miR-4772-3p had significantly shorter time to recurrence. A multivariate Cox regression model that included clinical-biological predictors of recurrence (tumor site, CEA) and levels of expression of miR-4772-3p showed that patients with low miR-4772-3p had a 5.48-fold higher recurrence risk (sensitivity 78.6%, specificity 77.1%, AUC 0.72 (95% CI: 0.59–0.85, *P* = 0.001); low miR-4772-3p and high CEA, but not tumor size, were significantly related to a higher risk of death. The only significant difference between the recurrence rates and miR-4772-3p was seen in stage III patients that received adjuvant FOLFOX: patients with low miR-4772-3p had higher recurrence rates than patients with high miR-4772-3p. The authors concluded that miR-4772-3p may serve as a biomarker for predicting tumor response to adjuvant FOLFOX chemotherapy, but it was unclear if it was a real predictive biomarker for chemo response or only a prognostic marker for tumor recurrence.

In another approach, Kjersem et al. [58] tested the hypothesis that plasma miRNAs can predict clinical outcomes in patients treated with first-line FOLFOX. Plasma samples from 24 patients (12 responders and 12 non-responders) with mCRC were examined at baseline and after four cycles of chemotherapy in search of differentially expressed miRs between the two groups with different clinical outcomes. The most significant miRNAs previously identified were validated on a cohort of 150 patients. Three of these, miRNA-106a, miR-484 and miR-130b were found to be upregulated in non-responders, having a significantly differential expression at baseline. Overexpression of miR-326, miR-27b, miR-148a was associated with low PFS, while miR-326 was associated with low OS. Nevertheless, after four cycles of chemotherapy, none of these miRNAs were statistically significant differentially expressed with regards to the outcome.

Zhang et al. [59], also evaluated the predictive role of circulating miRNAs for chemotherapy response in a study that included both screening and validation assessment, on the serum samples of 253 patients (80 in the screening and training phase, 173 in the validation phase) with stage III-IV CRC treated with mFOLFOX6. Their analysis pointed out a panel of five serum miRNAs (miR-20a, miR-130, miR-145, miR-216, miR-372) significantly differentially expressed between chemosensitive and chemoresistant patients, which could be considered as a biomarker for predicting the chemosensitivity of CRC. This set of five miRNAs was associated with response prediction and could be used as serum biomarkers for chemotherapy response in patients receiving oxaliplatin-based chemotherapy (92% sensitivity and 88% specificity).

In another study, Chen J et al. [60] wanted to determine if dynamic monitoring of miR-155, miR-200c, and miR-210 has a role in CRC diagnosis, prognosis and can predict chemoresistance. In an analysis performed on both serum samples (15 CRC stage III compared to 20 healthy donors) and CRC and healthy adjacent tissues (15 stage III patients), they observed a significant increase of the three miRNAs when compared to healthy counterparts. They assessed levels of miR-155, miR-200c and miR-210 on serum samples from patients with stage III CRC harvested at different time points after surgery and chemotherapy with mFOLFOX6+ Cetuximab treatment, for three years: three months after surgery and chemotherapy the serum levels of patients without disease recurrence were significantly lower and 12 months after the treatment they normalized. On the contrary, in patients with disease recurrence, the levels of the three miRNAs have significantly dropped down at six to 12 months after the end of treatment but were maintained elevated and rose again at 12–18 months after treatment, before the diagnosis of disease recurrence. When treatment with Avastin + 5-FU was administered in the patients with disease recurrence (*n* = 6), the levels of miR-155 were maintained low in responders while in non-responders were significantly elevated when compared to miR-200c and miR-210. The authors concluded that a new increasing or a sustained increase of post-therapy miR-155 serum levels predict chemoresistance, while elevated levels or re-elevation of miR-155, miR-200c and miR-210 witness about disease recurrence and represent a negative prognostic factor. Maintaining the same line, Shivapurkar et al. [61] explored if certain circulating miRNAs could be predictors of clinical evolution in mCRC patients enrolled in a phase two clinical trial, with Sunitinib and Capecitabineas 1^st^ line treatment in this setting. Following an exploratory screening test including 380 serum miRNAs, miR-296 proved to have a significant correlation with patients’ clinical outcomes. Eight patients were included; one had an undetectable baseline and after therapy levels of miR-296, so only seven were considered for the analyses. Patients whose levels of miR-296 decreased had a lower OS and unfavorable clinical outcomes compared to the patients with an increase in serum miR-296 and opposite clinical outcomes. The expression of miR-296 is progressively lost during tumor progression and it correlates with evolution to metastatic disease. Such as the patients with a low serum level of microRNA after four weeks represented a tumor type with a more aggressive phenotype and higher potential of invasion and metastasis.

In order to identify whole blood miRs with prognostic value in mCRC, Schou et al. [62] performed a study including patients treated in the third-line setting with Irinotecan and Cetuximab in a phase two prospective study. They isolated 738 pre-therapy miRNAs in whole blood samples of 138 patients, out of which, six miRNAs were associated with shorter OS: miR-345, miR-143, miR-34a, miR-628-5p, miR-886-3p, miR-324-3p. Among these, miR-345was proven to have the strongest prognostic value, significant for all patients, including the wild-type KRAS population. Higher expression of miR-345 was significantly associated with a lack of response to Irinotecan+ Cetuximab treatment in these patients.

Exploring the miRNAs targeting endothelial cells, it was also of interest to identify predictive markers for chemotherapy response in CRC. MiR-126 is expressed in endothelial cells and is essential in maintaining the integrity of blood vessels. MiR-126 is a tumor suppressor miRNA, usually downregulated in cancer. It targets VEGF-A, but also EGFL7 which is upregulated in sites with pathological angiogenesis. In a study performed in 2012, Hansen et al. [63] showed that in patients with mCRC treated with CapeOx as first-line chemotherapy, high expression of miR-126 in CRC tissue (n=89 patients) was significantly related to tumor response (partial response + complete response), with a positive predictive value of 90% and a negative predictive value of 71%. The median PFS in CRC patients with high expression of miR-126 was 11.5 months versus 6 months in patients presenting low miR-126 expression (*p* < 0.0001). Taking into account their previous results, Hansen et al. [64], performed a second study, aiming to investigate the prognostic value of miR-126 in mCRC and to establish a relationship between the efficacy of chemotherapy plus Bevacizumab and the expression of EGFL7. They included 249 patients from a phase three prospective multicentric clinical trial with mCRC that received as first-line treatment, chemotherapy with FOLFOX/XELOX or FOLFIRI/XELIRI, both regimens combined with Bevacizumab (six or nine cycles), with primary endpoint response rate; samples were available for 230 patients, with blood samples for 222 patients and adequate tissue samples for 169 patients. The only result with statistical significance pointed out that high tumor expression of miR-126 was related to a better PFS. The relationship between EGFL7 and response rates was only suggested (low EGFL7 in responding patients), while the expression of miR-126 did not correlate with EGFL7 tissue invasive front expression.

Furthermore, Hansen’s group continued their research [65] and analyzed the predictive value of circulating miR-126 in mCRC patients treated with first-line CapeOx plus Bevacizumab. The main objectives focused on RR (RECIST criteria) and PFS. A total of 68 patients were included and plasma samples were collected at baseline, three weeks after the treatment initiation and at disease progression. The results showed that dynamic changes in the levels of miR-126 during treatment were significantly predictive for tumor response (increase in plasmatic miR-126 in non-responding patients and decrease in responders, *p* = 0.0001). On the other hand, patients with decreasing miR-126 levels during chemotherapy had a marginally significantly better PFS (*p* = 0.07).

Ulivi and colleagues [66] also analyzed the role of circulating miRNAs in predicting the clinical outcome of 52 mCRC patients treated with a Bevacizumab-containing regimen (FOLFOX or FOLFIRI) within the ITACa clinical trial. Firstly, they analyzed the association between baseline circulating miRNAs and the clinical pathological characteristics of the patients and found hsa-miR-199a-5p, hsa-miR-335-5p and hsa-miR-520d-3p to be significantly upregulated in left-sided versus right-sided tumors and hsa-miR-21-5p and hsa-miR-221-3p significantly associated with the RAS mutational status. Further on, they identified baseline levels of circulating hsa-miR-20b-5p, hsa-miR-29b-3p and hsa-miR-155-5p as being significantly associated with PFS and OS in multivariate analysis. Also, an increase in hsa-miR-155-5p at the first evaluation of the treatment response was significantly associated with shorter PFS and OS. However, because of the small number of patients and the lack of a control arm, it is unclear whether miRNAs had a prognostic or a predictive value for the response to treatment.

Previous studies showed that miR-34a was down-regulated in multiple malignant tumors, CRC included. Sun et al. [67] investigated the role of miR-34a in resistance to L-OHP chemotherapy on CRC cells. Thirty CRC patients that underwent potentially curative surgery and were treated with L-OHP based adjuvant chemotherapy were included in the study. The expression of miR-34a was tested at baseline on CRC tissues but also on patients’ blood samples after L-OHP-based treatment. The expression of miR-34a was significantly reduced after exposure to L-OHP and inversely correlated with TGF beta and SMAD4 expression. The analysis of CRC cell lines resistant to L-OHP also revealed a suppression of miR-34a expression and an activated TGF beta/SMAD pathway. Moreover, the authors have proved that miR-34a is directly involved in regulating of SMAD4 expression and inhibition of TGF-beta and that activation of macroautophagy contributes to L-OHP treatment resistance. The study suggested that by suppressing miR-34a through TGF beta/SMAD4 pathway, the activation of macroautophagy could be a protective mechanism against L-OHP induced cellular death.

Alteration of miRNA expression in CRC, through the prism of single nucleotide polymorphisms (SNPs), was evaluated by Lin et al. [68] on 1097 CRC patients (741 in the training set and 346 in the replication set) in order to demonstrate a possible effect of SNPs in miRNAs-encoding genes on the prognosis of CRC patients treated with fluoropyrimidine-based chemotherapy. Out of the 41 SNPs tested in 26 miR related genes, *microRNA-608 rs 4919510* was associated with a higher risk of disease recurrence and death, while *microRNA-219 rs 213210* was associated with increased risk of death in patients with stage III disease treated with 5FU-based adjuvant chemotherapy. Furthermore, patients carrying both variant genotypes of these two SNPs had a 5.6-fold increased risk of death. Based on Lin et al.’s research, Pardini et al. [69] studied the effects of SNPs on blood samples obtained from 1083 CRC patients of any stage (I-IV) in the Czech Republic. None of the SNPs were significantly associated with OS and event-free survival (EFS) in all 1083 patients, but after stratification according to administration of 5-FU- based chemotherapy, the authors observed that chemotherapy-treated patients that exhibited the variant T allele of rs213210 in microR-219-1 had lower OS and event-free survival (EFS) than patients with a wild-type genotype, meaning a higher risk of disease recurrence and death. After further stratification of patients that had adjuvant 5-FU based chemotherapy according to disease stage, only patients with stage III disease that exhibited the variant G allele of rs4919510 (in miR-608) had a lower risk of disease recurrence than wild-type patients.

Polymorphisms of the miRNAs’ precursors alter either the expression of the miRNAs, its binding complementarities to mRNA targets, or both possibilities. Chen et al. [70] studied the predictive role of some polymorphisms in miRNAs precursors for Capecitabine-based chemotherapy efficacy. A lot of 274 advanced CRC patients who had first-line chemotherapy for metastatic disease with CapeOx and no prior chemotherapy regimen in adjuvant setting were included and blood samples and germline DNA were provided before the start of the chemotherapy. They detected six polymorphisms and identified the rs7911488 T>C polymorphism in pre-miR-1307 as being significantly associated with Capecitabine’s efficacy. Response rates to Capecitabine in patients with TT (homozygous for the T allele), TC (heterozygous) si CC (homozygous for the C allele) genotypes were 44.35% (55/124), 51.33% (58/113), and 24.32% (9/37). The CC homozygotes had significantly lower response rates compared to the TT homozygotes. In patients with C-allele, the resistance to Capecitabine was related to miR-1307-3p downregulation. In vitro and in vivo studies also proved that CRC cells with C allele rs7911488 are resistant to 5-FU. In another study, Boni et al. [71] analyzed the relationship between polymorphisms in miRNA-containing genomic regions (pri-miR, pre-miR) or genes involved in miRNAs biogenesis and clinical outcome of mCRC patients treated with 5-FU + Irinotecan. Peripheral blood from 61 patients was included, and 18 SNPs were studied. Among these, SNP rs 7372209 located in pri-miR-26a-1 was significantly associated with tumor response and time to disease progression. The genotypes CC (homozygous for the C allele) and CT (heterozygous) were favorable compared to TT (homozygous for the T allele) variant. SNP rs 1834306 located in pri-miR-100 was significantly correlated with a longer time to progression. The authors concluded that miR polymorphisms may represent predictive biomarkers of clinical outcome in advanced CRC patients treated with 5-FU and Irinotecan.

Extracellular miRNAs have two main sources: the miRNAs that cofractionate with Agonaut2 (Ago2) protein complexes which are thought to originate mainly from dead cells and the miRNAs that are released through extracellular vesicles (EV), mainly from viable cells and engaged in intracellular communication [72]. The authors hypothesized that by measuring levels of Ago2 miRNAs and EV miRNAs during chemotherapy, they would be able to assess the response to antitumor therapy. In vitro analysis showed that both Ago2-miRNAs and EV-miRNAs were released in the extracellular medium during 5-FU treatment, both through passive export due to cytolysis and active export from viable cells as a result of cellular stress. Out of the three miRNAs selected due to commonly expression in CRC tissues, miR-21, miR-31 and miR-200c, only Ago2-miR-21 showed the possibility of active release from viable cells, unexpectedly, in addition to its release through cytolysis. They further found that it was highly expressed in primary and metastatic CRC tissues and that it was released mainly by cytolysis in the extracellular medium. Afterward, they showed that high expression of Ago2-miR-21 and Ago2-miR-200c in advanced CRC patients could be used as biomarkers for treatment response.

A summary of all studies presented above, regarding the potential predictive role of miRNAs is provided in Table 2.

## 7. Limitations for Implementing miRNAs as a Prediction Tool for Chemoresponse into Daily Practice

Firstly, there are not enough clinical studies to draw a robust conclusion on the role of miRNAs as biomarkers in CRC. The majority of the above-presented studies confirmed the diagnostic and/or prognostic role of specific miRNAs in CRC patients on tissues or blood/serum/plasma, but most of the studies of prediction were performed on in vitro and in vivo models. There is, however, a limited number of studies that tested the chemo-prediction potential of miRNAs in patients that had response evaluations by radiological assays and the clinical response was correlated with the baseline level of a specific circulating miRNA or with dynamic changes in the levels of circulating miRNAs during systemic treatment.

## 8. Challenges in the Therapeutic Use

Considering the role of miRNAs as modulators of protein-coding genes, suppressing the oncogenic miRNAs or substituting the tumor-suppressor-deficient miRNAs represents the premises of research centered on introducing miRNA-based cancer treatments so that benefits are drawn not only from their diagnostic, prognostic and predictive value but also from modulating their activities in order to achieve antitumor activity at reduced costs and possibly with no adverse events. Current anticancer treatments are mainly involved in antiproliferative and proapoptotic pathways or modulate the immune responses generated by the presence of the tumors. The tumor phenotype can be controlled by modulating microRNAs expressions and this can be the basis of new successful approaches in cancer therapy [47].

There are several therapeutic approaches based on miRNA modulation and various ways to use the miRNA therapeutics, the vast majority being validated so far only by in vitro and in vivo studies. Firstly, the miRNA inhibitors oriented towards the oncogenic miRNAs: complementary single-stranded oligonucleotides that form, by binding to the miRNA, an unknown conformation leading to miRNA’s exclusion from the RISC complex where the miRNAs are typically processed; among the miRNA inhibitors, we mention AMOs (antisense antimiR oligonucleotides), LNAs (locked nucleic acids), antagomiRs, antimiRs, miRNA sponges, microRNA masks. Secondly, the miRNA mimetic agents (miRNA mimics), that use synthetic miRNA-like molecules that replace or substitute the lost tumor suppressor miRNA and can be loaded into the RISC complex and function as a miRNA, inhibiting the target mRNA. Besides the two mentioned above, the SMIRs (small molecules inhibitors of microRNAs) inhibit miRNAs biogenesis or interfere between miRNAs and their targets. A big challenge is to inhibit tumoral miRNAs’ and exosomes’ secretion, therefore suppressing the interaction between the tumor microenvironment’s components: cancer cells, immune cells, etc, as it is well known that miRNAs are transported through extracellular vesicles either between different cell types within the tumor microenvironment or from the primary tumor to metastatic sites [73].

Epithelial to mesenchymal transition (EMT) is a biological process by which epithelial cells lose their intercellular adhesions, gain migratory and invasive capabilities, become mesenchymal cells and so, they enter the metastatic cascade, which is coordinated by multiple signaling pathways. MiRNAs are among the most important regulators of the EMT and so, they may become critical therapeutic agents to block EMT and, therefore, invasion and metastasis [74]. MiR-625 is an inhibitor of metastasis, downregulated in colorectal cancer, gastric cancer, and hepatocellular carcinoma. Lou X, et al. studied the involvement of miR-625 in colorectal cancer and found that its decreased expression was associated with lymph node involvement, liver metastasis, poor overall survival and unfavorable prognosis [75]. In vitro and in vivo studies have shown that ectopic miRNA expression inhibited the invasion and migration of colorectal cancer cell lines. Taking these into consideration, restoring their deficient expression could represent a possible therapeutic approach.

Despite the promising in vitro and in vivo results of the miRNA therapeutics, many challenges and questions need accurate answers that lack when it comes to the mechanisms that trigger miRNAs’ regulation and most of these are probably to be found in the tumor microenvironment. The fact that the same microRNA can have opposite effects on different or multiple targets is well known. There are still significant uncertainties regarding the proper formulations in order to obtain efficient and specific methods of delivery, as well as an increased in vivo stability. Nonetheless, their profile of adverse events needs to be better understood and the dosage established in order to obtain effective treatments with tolerable adverse events.

## 9. Conclusions

The existence of predictive markers for tumor response to therapies allows a better selection of available treatment options, reduces unnecessary treatments, toxicities, costs, by allowing an adequate patient selection. Up to date research did not succeed in offering validated predictive biomarkers for systemic treatment response in mCRC. Regarding the above-mentioned standard drugs administered according to international guidelines, validated predictive factors exist only for the anti-EGFR targeted treatments and for immunotherapy used in refractory cases. Proposed predictive factors for chemotherapy or targeted treatment response are represented by gene expression, polymorphisms or mutations of genes involved in drug activity or levels of enzymes involved in drug metabolism or activity. Clinical features (PS), factors related to the natural history of the disease, and laboratory parameters, such as tumor markers or markers of baseline inflammation, are also among the proposed predictive factors. Treatment-related toxicity, either clinical or seen on the blood works often represents a surrogate marker for treatment efficacy.

In this context, the emergence of novel biomarkers such as miRNAs, still under investigation, could represent the beginning of a paradigm shift in the management of mCRC; they can offer important molecular data, are stable, require minimum volumes of blood; although expensive when compared to currently proposed markers, the actual costs could be reduced by better choice of treatments.

## Figures and Tables

**Table 1 ijms-21-02089-t001:** Validated and proposed predictive factors for response to systemic treatments in metastatic colorectal cancer.

Drug	Treatment Line in mCRC	Validated Predictive Factors	Proposed Predictive Factors			
			Tumor-Related	Clinical	Disease-Related	Laboratory
**5-FU**	**any**	**none**	**Response:** combined low tumor levels of TS, TP, and DPD [19] **Lack of response:** high tumor TS levels/specific TS polymorphisms (TSER*3/TSER*3) [16] high tumor TP levels [18]	**Lack of response:** performance status ≥ 2 [20]	**Response:** rectal primary, lung only/nodal metastases [20] **Lack of response:** number of metastatic sites ≥ 2 [20] presence of peritoneal carcinomatosis or liver metastases [20]	**Lack of response:** WBC ≥ 10 × 10^9^ /L [20] Hemoglobin < 11 × 109/L [20] Platelets ≥400 × 109 /L [20] Alkaline phosphatase ≥ 300 U/L [20]
**Capecitabine**	**any**	**none**	**Response:** low TS levels/specific TS polymorphisms [21] low DPD levels [22]	**Response:** hand-foot syndrome [13]		
**Irinotecan**	**any**	**none**	**Response:** high TOPO1 gene expression [23]	**Response:** performance status 0–1 [25] diarrhea at 1st cycle [24]	**Response:** time from diagnosis < 9 months [24] number of organs involved = 1 [24]	**Response:** normal baseline hemoglobin [24] G3/G4 neutropenia at first cycle [24]
**Oxaliplatin**	**any**	**none**	**Response:** low intratumoral ERCC-1 [8,12] low intratumoral TS [26]	**Lack of response:** ECOG ≥2 [26]	**Lack of response:** prior chemotherapy regimens ≥ 3 [26]	**Lack of response:** baseline Hb < 10 g/dL [26]
**Trifluridine-tipiracil**	**3rd, 4th**	**none**				**Response:** neutropenia after 1st cycle [27]
**Cetuximab, Panitumumab**	**any**	**Response:** RAS wild-type [12] Left primary tumor [12]	**Lack of response:** BRAF mutations (especially in ≥2nd line) [36,37] Amplifications of KRAS, HER2, MET PIK3CA exon 20 mutations, loss of PTEN [11] Increased TGF alpha Amphiregulin, epiregulin suppression [37]	**Response:** skin toxicity during treatment [40,41]		**Response:** Hypomagnesemia [37]
**Bevacizumab**	**1st, 2nd**	**none**	**Response:** VEGF gene polymorphism 1154 (G/G) VEGF gene polymorphism 634 was (G/G) [30]	**Response:** treatment-induced arterial hypertension [29]		**Response:** pre-therapy low SII indices pretherapy low NLR, PLR [30] abnormal baseline CA 19-9
**Aflibercept**	**2nd line**	**none**				**Response:** High baseline plasmatic IL8 [42]
**Ramucirumab**	**2nd line**	**none**	**Response:** High plasmaticVEGF-D expression [43]			
**Regorafenib**	**3rd line**	**none**		**Response:** -hand-foot skin reaction; lung nodule cavitation [44]		
**Larotrectinib/Entrectinib**	**refractory tumors**	**Response:** **NTRK gene fusion**				
**Pembrolizumab, Nivolumab, Ipilimumab**	**refractory tumors**	**Response:** **MSI-H, Dmmr** [46]				

**Table 2 ijms-21-02089-t002:** Studies for predictive microRNAs (miRNAs) associated to systemic treatments response in colorectal cancer (CRC).

Chemotherapy	CRC Stage	microRNA	Validation Study for the miRNAs Role in			Ref.
			Diagnosis	Prognosis	Prediction	
**5-FU**	**I-IV**	**miR-196b-5p**	**tissue** *n* = 20 CRC **serum, exosomes** *n* = 150 CRC *n* = 90 healthy	**tissue** *n* = 90 CRC **serum, exosomes** *n* = 150 CRC *n* = 90 healthy	***in vitro*, in vivo** models	Ren et al. [52]
	**IV**	**Ago2-miR-21,** **Ago2-miR-200c**	**plasma** *n* = 40 + 20 CRC		**plasma** *n* = 40 + 20 CRC	Fuji T et al. [72]
**5-FU/Capecitabine**	**I-IV**	**miR-608 rs 4919510,** **miR-219 rs 213210**		**peripheral blood** *n* = 356 CRC		Lin et al. [68]
**5-FU-based**	**I-IV**	**miR-429**	**tissue** *n* = 78 CRC **serum** *n* = 45 CRC *n* = 45 healthy	**tissue** *n* = 78 CRC **serum** *n* = 45 CRC *n* = 45 healthy	**tissue** *n* = 116 CRC (stage IV)	Dong et al. [51]
		**miR-608 rs 4919510, miR-219 rs 213210**		**peripheral blood** *n* = 1083 CRC		Pardini et al. [69]
**5-FU + Irinotecan**	**IV**	**pri-miR 26a-1 rs 7372209,** **pri-miR-100 rs 1834306**		**peripheral blood** *n* = 61 CRC		Boni et al. [71]
**5-FU + Avastin**	**recurrence after stage III adjuvant treatment**	**miR-155**		**serum** *n* = 6 CRC		Chen et al. [60]
**Oxaliplatin**	**not specified**	**miR-135b**	**serum** *n* = 25 CRC *n* = 25 healthy		***in vitro*, in vivo** models	Qin et al. [53]
	**A-D (Duke)**	**miR-143**	**tissue** *n* = 62 CRC **plasma** *n* = 41 CRC *n* = 10 healthy	**tissue** *n* = 62 CRC **plasma** *n* = 41 CRC *n* = 10 healthy	***in vitro***	Qian et al. [54]
	**any**	**miR-378**	**tissue** *n* = 20 CRC **serum** *n* = 37 CRC *n* = 14 healthy	***in vitro*, in vivo** models	***in vitro*, in vivo** models	Wang et al. [55]
**Oxaliplatin-based**	**not specified (adjuvant)**	**miR-34a**			**plasma** *n* = 30 CRC **CRC cell lines**	Sun C et al. [67]
**FOLFOX**	**II-III**	**miR-4772-3p**		**serum** *n* = 84 CRC	**serum** *n* = 84 CRC	Liu et al. [57]
	**III-IV**	**miR-20a, miR-130, miR-145, miR-216, miR-372**			**serum** *n* = 173 CRC	Zhang J et al. [59]
	**IV**	**miR-19a**			**serum** *n* = 72 CRC	Chen Q et al. [56]
	**IV**	**miR-326, miR-27b, miR-148a, miR-106a, miR-484, miR-130b**		**miR-326, miR-27b, miR-148a** **plasma** *n* = 150 CRC	**miR-106a, miR-484, miR-130b** **plasma** *n* = 150 CRC	Kjersem et al. [58]
**CapeOx**	**IV**	**miR-126**		**tissue** *n* = 89 CRC		Hansen et al. 2012 [63]
		**Rs7911488 miR-1307-3p**			**blood** *n* = 274 CRC	Chen Q et al. [70]
**CapeOx + Bevacizumab**	**IV**	**miR-126**		**plasma** *n* = 68 CRC		Hansen et al. 2015 [65]
**Capecitabine + Sunitinib**	**IV**	**miR-296**		**serum** *n* = 7 CRC		Shivapurkar et al. [61]
**mFOLFOX6 + Cetuximab**	**III**	**miR-155, miR-210, miR-200c**	**serum** *n* = 15 CRC *n* = 20 healthy **tissue** *n* = 15 CRC	**serum** *n* = 15 CRC		Chen et al. [60]
**FOLFOX/XELOX/FOLFIRI/XELIRI + Bevacizumab**	**IV**	**miR-126**		**tissue** *n* = 169 CRC		Hansen et al. 2013 [64]
**FOLFOX/FOLFIRI + Bevacizumab**	**IV**	**hsa-miR-20b-5p, hsa-miR-29b-3p and hsa-miR-155-5p**		**plasma** *n* = 52 CRC		Ulivi et al. [66]
**Irinotecan + Cetuximab**	**IV**	**miR-345**		**whole blood** *n* = 138 CRC		Schou et al. [62]

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
