# Peer review of "Current and New Predictors for Treatment Response in Metastatic Colorectal Cancer. The Role of Circulating miRNAs as Biomarkers"

_ijms, 2020, doi:10.3390/ijms21062089_

Round 1

Reviewer 1 Report

This is a nice review dealing with the “current and new predictors for treatment response in metastatic colorectal cancer” giving emphasis on “the role of circulating miRNAs as biomarkers in mCRC”. In my opinion, this review worth publication after taking into account the suggestions given below.

I consider this review as an effort to target concurrently two tasks, namely the role of predictive factors in mCRC as a whole, and the role of miRNAs in more details. However, as there are recent relevant detailed reviews (Julien Taieb, Andreas Jung, Andrea Sartore‑Bianchi, Marc Peeters, Jenny Seligmann, Aziz Zaanan, Peter Burdon, Clara Montagut, Pierre Laurent‑Puig. The Evolving Biomarker Landscape for Treatment Selection in Metastatic Colorectal Cancer. Drugs 2019;79:1375–1394 https://doi.org/10.1007/s40265-019-01165-2), I propose the authors to deal exclusively with the role of miRNAs as predictors of mCRC, as the relevant current reviews are dealing with CRC in general (Óscar Rapado-González, Ana Álvarez-Castro, Rafael López-López, José Iglesias-Canle, María Mercedes Suárez-Cunqueiro, Laura Muinelo-Romay. Circulating microRNAs as Promising Biomarkers in Colorectal Cancer. Cancers 2019,11,898; doi:10.3390/cancers11070898). In this regard, following an introductory part referring to the available and emerging predictive biomarkers, the text might be focused on the available data regarding the role of miRNAs in mCRC. In my opinion, this could be more useful to the readers.

Author Response

Dear Reviewer,

We thank you for your pertinent observations regarding our study and we consider that by addressing your comments we significantly improved the quality of our manuscript entitled “Current and new predictors for treatment response in metastatic colorectal cancer. The role of circulating miRNAs as biomarkers”.

Please find bellow, a point by point response to your comments.

With the consideration that we answered the suggestions and comments, we are looking forward for your decision regarding our work.

Comment:

I consider this review as an effort to target concurrently two tasks, namely the role of predictive factors in mCRC as a whole, and the role of miRNAs in more details. However, as there are recent relevant detailed reviews (Julien Taieb, Andreas Jung, Andrea Sartore‑Bianchi, Marc Peeters, Jenny Seligmann, Aziz Zaanan, Peter Burdon, Clara Montagut, Pierre Laurent‑Puig. The Evolving Biomarker Landscape for Treatment Selection in Metastatic Colorectal Cancer. Drugs 2019;79:1375–1394 https://doi.org/10.1007/s40265-019-01165-2) ), I propose the authors to deal exclusively with the role of miRNAs as predictors of mCRC.

Answer:

Our efforts were centered on identifying non-invasive, predictive biomarkers for response to treatments, both inexpensive and available for every physician, accessible from the baseline information mandatory for every CRC patient before chemotherapy start, but also, more expensive and specific assays like enzymatic levels, polymorphisms, other molecular markers and finally, the circulating miRNAs.

We studied the review article suggested (Taieb et al., 2019) and found it very relevant and comprehensive regarding enzymatic, genetic and molecular biomarkers for treatment response, in the light of the existing ESMO, ASCO, CAP Guidelines and introduced it in our text (line 73). Besides the validated tumor sidedness, they did not discuss, however, the value of baseline clinical evaluation, baseline investigations and toxicities (e.g. clinical examination- the performance status, CT scans- sites of metastatic disease and no of metastatic sites, tumor sidedness, history of the disease- time from diagnosis, basic blood works- hemoleucograms with inflammatory indices, alkaline phosphatase, blood tumor markers and their dynamics, toxicities as surrogate markers for treatment response). Indeed, these factors are still not validated, but until tumor biology is better deciphered, they could represent inexpensive alternatives to more complex, but equally not validated assays.

We tried to shorten the text as much as we could without losing relevant information regarding the predictive biomarkers presented in the first part (clinical, laboratory, toxicities, enzymatic).

These were our premises, but if you consider that our work will be of better use and value for the readers if we significantly reduced the first part and exclude Table I from the manuscript, then we will be happy to proceed accordingly.

Comment:

I propose the authors to deal exclusively with the role of miRNAs as predictors of mCRC, as the relevant current reviews are dealing with CRC in general (Óscar Rapado-González, Ana Álvarez-Castro, Rafael López-López, José Iglesias-Canle, María Mercedes Suárez-Cunqueiro, Laura Muinelo-Romay. Circulating microRNAs as Promising Biomarkers in Colorectal Cancer. Cancers 2019,11,898; doi:10.3390/cancers11070898).

Answer:

We found the suggested review article to be very relevant for the topic of circulating miRNAs as biomarkers in CRC and therefore, introduced it as a reference work in our text (line 65). Our focus was indeed set on the predictive value of miRNAs and we only included studies that had this objective. However, a limited number of the included studies dealt exclusively with the predictive value for treatment response, while the majority of them also intended to establish their diagnostic and/or prognostic role. Table II summarizes the characteristics and the objectives of every study included in this review.

Also, we added an additional relevant study (Ulivi P et al, 2018), that was initially omitted from the manuscript submitted (lines 625 and in Table 2).

Additionally, in response to the 2nd reviewer’s suggestions, we added some paragraphs in the introduction (lines 65-74), in order to better address the research question and aims and introduced a ‘Methods’ section (lines 80-98).

The studies regarding miRNAs included all CRC stages, not just metastatic disease due to low number of studies for stage IV-only (as presented in the ‘methods’ section) and we modified the subtitles accordingly and the title of the Table II, as it was initially wrongly defined.

As a general remark, we inserted new references where they were required and adjusted in consequence the bibliography, and also verified once again the English language, all over the text.

Reviewer 2 Report

the paper is a narrative review aimed to indentify non invasive markers of treatment response in patients with metastatic colorectal cancer.

Identifing such markers could improve survival in such patients and help to select therapies adequate to the patient.

Despite the scientifical soundness there is a lack in methods. AA should describe the research question, the search strategy, the sources and databases consulted, the quality assessment of studies, study design and inclusion and exclusion criteria. Where possible, predictivity indicators (sensibility, sensitivity, positive and negative predictive values and so on) should be extracted. IN the present way the review could be biased and we have no tools to detect such bias

Author Response

Dear Reviewer,

We thank you for your pertinent observations regarding our study and we consider that by addressing your suggestions we significantly improved the quality of our manuscript entitled “Current and new predictors for treatment response in metastatic colorectal cancer. The role of circulating miRNAs as biomarkers”. Please find bellow the responses to your suggestions.

With the consideration that we completely answered all the suggestions, we are looking forward for your decision regarding our work.

Comments:

Despite the scientifical soundness there is a lack in methods. AA should describe the research question, the search strategy, the sources and databases consulted, the quality assessment of studies, study design and inclusion and exclusion criteria. Where possible, predictivity indicators (sensibility, sensitivity, positive and negative predictive values and so on) should be extracted. IN the present way the review could be biased and we have no tools to detect such bias.

Answer:

In the Introduction, we added a paragraph in order to better address the research question, as suggested (lines 69-78).

We also introduced the ‘Methods’ section as recommended (lines 84-107), where we detailed the search strategy, the sources and databases consulted, study design and inclusion and exclusion criteria. For each of the included studies we described in the text the predictivity indicators and we also mentioned this aspect in the ‘Methods’.

Round 2

Reviewer 1 Report

The authors have done a perfect work.

Reviewer 2 Report

I am satisfied by the AA answers, so recommend to publish the article in present form